# Unimproved source of drinking water and its associated factors: Multivariable analysis of Somalia Integrated Household Budget Survey (SIHBS 2022)

Omar Muhumed Maidhane[1], Omran Salih[1,2]*

1 School of Postgraduate Studies and Research, Amoud University, Borama, Somalia, 2 Institute of Systems Science, Durban University of Technology, Durban, South Africa

* omran.salih@amoud.edu.so

## Abstract

This study investigates the prevalence and factors associated with the use of unimproved drinking water sources in Somalia, a nation severely impacted by conflict, climate change, and a large nomadic population, thus hindering progress towards Sustainable Development Goal 6 (SDG 6). A cross-sectional analysis was conducted using data from 11,949 households from the 2022 Somalia Integrated Household Budget Survey (SIHBS). Logistic regression was employed to identify associations between unimproved water source usage and various individual and community-level factors. The results show significant geographical disparities, with region being a strong predictor. Surprisingly, urban households demonstrated a higher likelihood of using unimproved water sources compared to rural households (Adjusted Odds Ratio [AOR] = 2.067). At the household level, unimproved water use was more prevalent in households headed by divorced (AOR = 1.598) or never-married individuals (AOR = 1.262), and in those residing in permanent or semi-permanent housing (AOR = 1.385). Conversely, food-insecure households had lower odds of using unimproved water (AOR = 0.878). The use of unimproved drinking water in Somalia is shaped by a complex interaction of community and household factors. These findings highlight the urgent need for geographically targeted interventions that address regional inequalities, the specific challenges of urban water supply, and household vulnerabilities. Further research is recommended to explore the unexpected associations with urban living and food insecurity, informing context-specific strategies by Somali authorities and international partners to improve public health and accelerate progress towards SDG Target 6.1.

**Data availability statement:** The data used in this study is sourced from publicly available datasets, ensuring accessibility for verification and further research. The data can be found (https://microdata.nbs.gov.so/index.php/catalog/59).

**Funding:** The author(s) received no specific funding for this work.

**Competing interests:** The authors have declared that no competing interests exist.

## 1. Introduction

Access to safe drinking water is a fundamental human right and a critical component of Sustainable Development Goal 6 (SDG 6.1) [1,2]. The WHO/UNICEF Joint Monitoring Program (JMP) distinguishes between 'improved' sources (e.g., piped water, boreholes), designed to protect against contamination, and 'unimproved' sources (e.g., unprotected wells, surface water), which pose significant health risks [3,4]. Globally, while 73% of the population used safely managed drinking water in 2022, 2.2 billion people lacked such access, with Sub-Saharan Africa (SSA) facing the lowest coverage [4]. In East Africa, reliance on unimproved sources remains high; for instance, preliminary 2020 data suggested around 22% of Somali households used unimproved sources [5], compared to approximately 31% in Ethiopia [6].

Household reliance on unimproved water is shaped by individual factors like socio-economic status and education of the household-head [7–8], and community-factors such as urban/rural residence and regional disparities [5,10]. Consumption of unimproved water leads to waterborne diseases, including diarrheal diseases, a major cause of child mortality [11,12], and imposes significant time burdens, particularly on women and girls [13].

Somalia faces unique challenges due to prolonged conflict, climate shocks (droughts, floods), and a significant nomadic population, all hindering sustainable water infrastructure development [5,6]. While the 2020 Somalia Health and Demographic Survey (SHDS) provided initial insights [5], and the Somalia Integrated Household Budget Survey (SIHBS) 2022 offers the most recent nationally representative data, a comprehensive multivariable analysis identifying both individual/household and community/regional determinants of unimproved water source use in this context is currently lacking.

This study, therefore, aims to investigate the prevalence of unimproved drinking water source use and identify its associated individual and community factors among households in Somalia, utilizing data from the SIHBS 2022. By employing multivariable logistic regression, we seek to provide evidence to inform targeted policies and interventions to improve public health and accelerate progress towards SDG Target 6.1 in Somalia.

## 2. Materials and methods

### 2.1 Study design

This study employed a cross-sectional study design utilizing secondary data analysis of the Somalia Integrated Household Budget Survey (SIHBS) conducted in 2022 [14]. This design is appropriate for investigating the prevalence of unimproved drinking water source usage and identifying associated factors at a specific point in time.

### 2.2 Study setting

The study was conducted in Somalia; a country located in the Horn of Africa. The SIHBS 2022 dataset covers various geographical areas across the country, encompassing urban, rural, and nomadic populations, providing a comprehensive national

context for the analysis. Somalia faces significant challenges related to infrastructure, conflict, climate shocks, and population mobility, making it a critical setting for studying access to basic services like safe drinking water.

## 2.3 Data source

The data used in this study were derived from the 2022 Somalia Integrated Household Budget Survey (SIHBS) (DDI-SOM-SNBS-SIHBS-2022-V02) [14]. This nationally representative survey collects detailed information on household consumption, expenditure, income, demographics, housing conditions, and access to essential services such as water and sanitation, among other socio-economic indicators. The SIHBS 2022 employed a multi-stage stratified sampling design to ensure broad representativeness across national, regional, and possibly urban, rural, and nomadic populations. In the first stage, primary sampling units (PSUs) typically corresponding to enumeration areas (EAs) were selected. In the second stage, households were randomly selected within these PSUs. Households hierarchical sampling structure, where households are nested within communities or EAs, a multivariable analytical approach is appropriate. For the purposes of this study, the anonymized household-factors dataset from SIHBS 2022 was used to conduct the analysis.

Although the 2022 Somalia Integrated Household Budget Survey (SIHBS) provides a rich and nationally representative dataset, the use of secondary data entails certain limitations. The variables available for analysis were restricted to those collected during the original survey, which may not fully capture all dimensions of interest in the current study. Additionally, the data were collected for broader socioeconomic monitoring rather than exclusively for water and sanitation research, which may influence variable specificity and contextual detail. Nevertheless, the SIHBS employs standardized survey procedures, robust sampling methods, and quality assurance measures, enhancing the reliability and validity of the findings while allowing for generalizable insights at both household and community levels.

## 2.4 Study population

The study population, drawn from the Somalia SIHBS 2022, encompassed all private households across Somalia, including urban, rural, and nomadic populations, while excluding institutional populations. To ensure representativeness, the SIHBS 2022 utilized a multi-stage stratified random sampling design. This involved an initial selection of Primary Sampling Units (PSUs), typically Enumeration Areas (EAs), within each stratum (e.g., region, urban/rural/nomadic type), likely using probability proportional to size (PPS). Subsequently, a predetermined number of households were systematically or randomly selected from updated listings within these chosen PSUs. The SIHBS 2022 dataset initially provided records for 47,623 households before any study-specific processing. For the current analysis, this dataset underwent rigorous cleaning and preparation, during which households with missing information on the primary outcome variable (source of drinking water). The final analytical sample size for this study consisted of 11,949 households, which is the sample reflected in the descriptive statistics (Table 1) and used for all subsequent bivariate and multivariable analyses.

## 2.5 Study variables

**2.5.1 Dependent variable.** The dependent variable in this study is the use of unimproved drinking water sources. Based on the SIHBS classification, water sources are categorized as *improved* or *unimproved*. Unimproved sources include unprotected wells and springs, surface water, and deliveries from small tankers or carts—options generally lacking infrastructure to prevent contamination. In contrast, improved sources such as piped water, protected wells, boreholes, and rainwater collection are considered safer due to better protection from pollutants. While these classifications offer a useful framework, the study acknowledges potential variations in water quality that this binary categorization may not fully capture.

**2.5.2 Independent variables.** The study examined a range of factors at both the individual and community factorss. On the individual factors, variables included the age of the household head (grouped into categories), whether the

**Table 1. Univariate and Bivariate Analysis of individual-level Factors Associated with Drinking Water Source.**

| Variable | Category | Frequency(n) | Percentage% | Unimproved Water (%) | Improved Water (%) | Chi-square | df | P-value |
|---|---|---|---|---|---|---|---|---|
| Age of household head | <20 years | 2984 | 24.97 | 18.80 | 81.20 | **15.6342** | 5 | **0.008** |
| | 20-29 years | 1919 | 16.06 | 21.21 | 78.79 | | | |
| | 30-39 years | 3154 | 26.40 | 21.69 | 78.31 | | | |
| | 40-49 years | 1147 | 9.60 | 23.63 | 76.37 | | | |
| | 50-59 years | 928 | 7.77 | 19.61 | 80.39 | | | |
| | 60 + years | 1817 | 15.21 | 21.41 | 78.59 | | | |
| Sex of household head | Male | 5500 | 46.03 | 20.84 | 79.16 | **0.0079** | 1 | **0.929** |
| | Female | 6449 | 53.97 | 20.90 | 79.10 | | | |
| School Attendance | Yes | 6142 | 51.40 | 19.59 | 80.41 | **12.6473** | 1 | **0.000** |
| | No | 5807 | 48.60 | 22.23 | 77.77 | | | |
| Marital Status of household head | Married | 5612 | 46.97 | 23.27 | 76.73 | **44.4054** | 3 | **0.000** |
| | Divorced | 630 | 5.27 | 14.60 | 85.40 | | | |
| | Never married | 4888 | 40.91 | 19.31 | 80.69 | | | |
| | Widowed | 819 | 6.85 | 18.56 | 81.44 | | | |
| Housing type | Temporary/Informal/Basic | 3488 | 29.19 | 28.50 | 71.50 | **173.4392** | 1 | **0.000** |
| | Permanent/Formal | 8461 | 70.81 | 17.73 | 82.27 | | | |
| Lighting source category | Traditional/Other | 3918 | 32.79 | 29.89 | 70.11 | **286.8973** | 1 | **0.000** |
| | Modern/Improved | 8031 | 67.21 | 16.47 | 83.53 | | | |
| Household Ownership | Yes (Household member) | 10599 | 88.70 | 21.37 | 78.63 | **14.0817** | 1 | **0.000** |
| | No (non-household member) | 1350 | 11.30 | 16.96 | 83.04 | | | |
| IDP Status | Yes | 515 | 4.31 | 21.75 | 78.25 | **0.2498** | 1 | **0.617** |
| | No | 11434 | 95.69 | 20.83 | 79.17 | | | |
| Internet Use | Yes | 4384 | 36.69 | 17.63 | 82.37 | **44.0067** | 1 | **0.000** |
| | No | 7565 | 63.31 | 22.75 | 77.25 | | | |
| Electricity Access | Yes | 7441 | 62.27 | 16.01 | 83.99 | **282.7814** | 1 | **0.000** |
| | No | 4508 | 37.73 | 28.90 | 71.10 | | | |
| Labour Status | Yes (worked in past week) | 1570 | 13.14 | 20.45 | 79.55 | **0.1988** | 1 | **0.656** |
| | No (did not work) | 10379 | 86.86 | 20.94 | 79.06 | | | |
| Food Insecurity | Food Secure | 5471 | 45.79 | 19.47 | 80.53 | **12.0750** | 1 | **0.001** |
| | Food Insecure | 6478 | 54.21 | 22.06 | 77.94 | | | |

household head had ever attended school, the respondent's educational attainment (none, primary, secondary, or higher), the gender of the household head (male or female), and their marital status (married, divorced, widowed, or never married). At the community factors, the study considered the type of residence—whether the household was located in an urban, rural, or nomadic area as well as the region where the household was situated. The regions included were: Awdal, Waqooyi Galbeed, Togdheer, Sool, Sanaag, Bari, Nugaal, Mudug, Galgaduud, Hiraan, Middle Shabelle, Banadir, Bay, Bakool, Gedo, and Lower Juba.

## 2.6 Data management and statistical analysis

Data management involved preparing the SIHBS 2022 dataset, including addressing missing values for the outcome and key predictors, with cases having critical missing information excluded. Statistical analysis was conducted using software capable of handling complex survey data and multivariable modeling. Descriptive statistics (frequencies, percentages)

were first used to characterize the study population and determine the prevalence of unimproved water source usage. Bivariable associations between independent variables and the outcome were then explored using Chi-square tests.

The primary analytical approach was a two-level random intercept logistic regression model, accounting for the nesting of households (Level 1) within communities/EAs (Level 2). Multivariable analysis was essential to control for confounding and simultaneously assess the independent contributions of individual- and community-level factors to the outcome, ensuring that observed associations were not explained by other correlated variables. This approach also allowed for estimation of the Intraclass Correlation Coefficient (ICC) from the empty model to quantify variance attributable to community-level clustering, followed by sequential models incorporating individual factors, community factors, and finally the combined model. Adjusted Odds Ratios (AORs) with 95% Confidence Intervals (CIs) were calculated from the final model to quantify associations, with statistical significance defined as $p < 0.05$.

## 2.7 Novelty of the study

The novelty of this study lies in the context, dataset, and findings rather than the statistical method alone. First, it represents the first comprehensive multivariable analysis of the nationally representative SIHBS 2022 dataset, providing up-to-date evidence on household drinking water access in Somalia, a country where such nationally representative analyses remain scarce due to conflict and fragility. Second, by situating water access within the Somali context of ongoing conflict, displacement, and climate vulnerability, this study generates insights that differ from those reported in broader Sub-Saharan African settings. Third, the identification of counterintuitive associations such as urban households being more likely than rural households to rely on unimproved water sources, and food-insecure households showing lower odds of doing so, challenges prevailing assumptions and underscores the complex and context-specific nature of water access in Somalia. Finally, the study provides policy-relevant evidence, highlighting regional disparities and household vulnerabilities that can inform Somali authorities and international partners working toward Sustainable Development Goal 6.1. Together, these contributions establish the originality and added value of this research beyond the application of logistic regression, which is acknowledged as a common analytical tool.

## 2.8 Ethical considerations

This research utilized secondary data obtained from the Somalia Integrated Household Budget Survey (SIHBS) 2022. The data were anonymized before being accessed, thereby protecting the confidentiality of participating households and individuals. Informed consent was secured from all participants during the original data collection phase conducted by the relevant Somali authorities and their partners. Because this study relies exclusively on pre-existing, de-identified data, specific ethical approval for this secondary analysis was not necessary, however, all data use agreements and established ethical guidelines for secondary research were strictly adhered to.

## 3. Results

### 3.1 Univariate and bivariate analysis

Table 1 presents a univariate analysis of various individual and community-factors from the SIHBS 2022 dataset, likely describing the characteristics of households surveyed (N = 11,949). Demographically, household heads were relatively young, with the largest proportions being aged 30–39 years (26.40%) and under 20 years (24.97%). A slight majority of household heads were female (53.97%), and school attendance among them was almost evenly split, with 51.40% having attended school. In terms of marital status, nearly half (46.97%) were married, while a significant portion (40.91%) had never been married. Regarding household characteristics, a majority resided in permanent or formal housing (70.81%) and utilized modern or improved lighting sources (67.21%). Dwelling ownership by a household member was very high at 88.70%. A small minority (4.31%) identified as Internally Displaced Persons (IDPs). While over a third (36.69%) had internet

access, the majority (63.31%) did not. Conversely, a majority (62.27%) had access to electricity. In terms of economic status, a small percentage (13.14%) of household heads reported engaging in paid labor in the past week. Finally, a slight majority of the households (54.21%) were classified as food insecure. These characteristics provide a baseline understanding of the surveyed population, which can then be used to analyze factors associated with their source of drinking water.

Again Table 1 presents bivariate analysis of factors associated with drinking water sources using the SIHBS 2022. The Chi-square tests of independence were conducted to examine the association between various socio-demographic and economic characteristics and the source of household drinking water (categorized as improved or unimproved). The significance factors for all tests was set at $\alpha = 0.05$. A statistically significant association was observed between the age of the household head and the drinking water source ($\chi^2$=15.6342, $p$=0.008). Similarly, strong, statistically significant associations were found for housing type ($\chi^2(1)$ = 173.4392, $p$<0.001), lighting source ($\chi^2(1)$ = 286.8973, $p$<0.001), and electricity access ($\chi^2(1)$ = 282.7814, $p$<0.001), indicating that households with better infrastructure were significantly more likely to use improved water sources. Further, household ownership status demonstrated a significant relationship with water source ($\chi^2(1)$ = 14.0817, $p$ < 0.001). Access to and use of modern amenities and information, such as internet use ($\chi^2(1) = 0.001$) and school attendance ($\chi^2(1) = 0.001$), were also significantly associated with the utilization of improved drinking water. Marital status was another significant factor ($\chi^2(3) = 0.001$). Finally, food insecurity status was also found to be significantly associated with the source of drinking water ($\chi^2(1) = 0.001$). Conversely, no statistically significant association was found between the sex of the household head and the drinking water source ($\chi^2(1) = 0.929$). Similarly, IDP status ($\chi^2(1) = 0.617$) and labour status ($\chi^2(1) = 0.656$) did not show a significant relationship with the type of drinking water utilized by the household. These findings suggest that, within this sample, these particular characteristics are not primary differentiating factors for access to improve versus unimproved water sources. In summary, the bivariate analysis indicates that multiple socio-economic, demographic, and geographic factors are significantly correlated with the type of drinking water source available to households, while a few others, such as sex of household head, IDP status, and recent labour activity, do not appear to be significant predictors in this context.

The examination of community characteristics related to sources of drinking water as depicted in Table 2 shows that both residency of the individual and region of the individual are significantly correlated with the water source they use, i.e., whether it is improved or unimproved. Residence shows an extreme degree of association ($\chi^2 = 355.1553$, $df$ =2, P<0.0001). Rural and Nomadic populations have much higher dependence on unimproved water sources (29.09% and 32.36%) relative to urban populations where 15.48% use unimproved water and 84.52% use improved sources.

Likewise, region shows a very strong association with drinking water source ($\chi^2 = 2300$, df =16, P<0.0001). Considerable variability exists within regions. Banadir and Lower Shabelle are outliers with 100% and 99.12% improved water usage, respectively. However, as Gedo, Sanaag, Sool, and Bakool exhibit higher unimproved water usage at 46.90%, 46.51%, 43.22%, and 43.28%, respectively. Waqooyi Galbeed and Togdheer also command considerable proportions of unimproved water users (42.14% and 30.82%). This illustrates extreme inter-regional differences in the provision of improved drinking water.

The Distribution of Unimproved Drinking Water Among Somali Households using SIHBS 2022 in Fig 1 maps starkly illustrates significant regional disparities, with the highest prevalence of unimproved drinking water concentrated in Somalia's northwestern and northern regions, notably Woqooyi Galbeed (63.35%) and Sanaag (50.31%). Conversely, central and southeastern regions like Shabeellaha Hoose (2.23%) show much lower rates. This highlights an urgent need for targeted interventions and water infrastructure investment in the most affected areas to address critical public health concerns. The Spatial Distribution of Poverty Magnitude among Somali Households (SIHBS 2022) was analyzed using custom code, which is provided in S1.

## 3.2 Regression analysis

The multivariable logistic regression analysis reveals several key factors significantly associated with the unimproved source of drinking water, while others showed no statistically significant effect (Table 3). Place of residence and region

**Table 2. Univariate and Bivariate Analysis of community-factors Associated with Drinking Water Source.**

| Variable | Category | Frequency(n) | Percentage% | Unimproved Water (%) | Improved Water (%) | Chi-square | df | P-value |
|---|---|---|---|---|---|---|---|---|
| Residence | Rural | 3458 | 28.94 | 29.09 | 70.91 | **355.1553** | **2** | **0.000** |
| | Urban | 7462 | 62.45 | 15.48 | 84.52 | | | |
| | Nomadic | 1029 | 8.61 | 32.36 | 67.64 | | | |
| Region | Awdal | 673 | 5.63 | 10.10 | 89.90 | **2300** | **16** | **0.000** |
| | Bakool | 737 | 6.17 | 43.28 | 56.72 | | | |
| | Banadir | 857 | 7.17 | 0.00 | 100.00 | | | |
| | Bari | 703 | 5.88 | 9.53 | 90.47 | | | |
| | Bay | 622 | 5.21 | 11.90 | 88.10 | | | |
| | Galgaduud | 782 | 6.54 | 6.01 | 93.99 | | | |
| | Gedo | 597 | 5.00 | 46.90 | 53.10 | | | |
| | Hiraan | 663 | 5.55 | 10.56 | 89.44 | | | |
| | Lower Juba | 388 | 3.25 | 7.73 | 92.27 | | | |
| | Lower Shabelle | 568 | 4.75 | 0.88 | 99.12 | | | |
| | Waqooyi Galbeed | 1260 | 10.54 | 42.14 | 57.86 | | | |
| | Middle Shabelle | 680 | 5.69 | 6.47 | 93.53 | | | |
| | Mudug | 517 | 4.33 | 5.03 | 94.97 | | | |
| | Nugaal | 719 | 6.02 | 8.62 | 91.38 | | | |
| | Sanaag | 645 | 5.40 | 46.51 | 53.49 | | | |
| | Sool | 782 | 6.54 | 43.22 | 56.78 | | | |
| | Togdheer | 756 | 6.33 | 30.82 | 69.18 | | | |

stand out as particularly strong predictors. Households in Urban areas had significantly higher odds (AOR = 2.067, 95% CI: 1.807–2.364, p < 0.001) of experiencing the unimproved water sources compared to those in Rural areas; Nomadic residence was not significantly different from Rural. Regional disparities were pronounced: compared to the Banadir region, individuals in Bakool (AOR = 0.165), Gedo (AOR = 0.118), Waqooyi Galbeed (AOR = 0.139), Sanaag (AOR = 0.109), Sool (AOR = 0.129), and Togdheer (AOR = 0.247) all exhibited substantially and statistically significantly lower odds of the outcome (all p < 0.001), with their 95% CIs falling entirely below 1. Conversely, those in Lower Shabelle (AOR = 11.722, p < 0.001) had dramatically higher odds, over eleven times those in Banadir, although its wide 95% CI (4.679–29.365) suggests some imprecision in this large estimate. Galgaduud (AOR = 1.780, p = 0.004), Middle Shabelle (AOR = 1.652, p = 0.014), and Mudug (AOR = 2.081, p = 0.003) also showed significantly increased odds.

Regarding socio-demographic characteristics, Marital Status was significant: divorced individuals had 1.598 times the odds (95% CI: 1.234–2.071, p < 0.001) and never-married individuals had 1.262 times the odds (95% CI: 1.067–1.492, p = 0.006) of the unimproved water sources compared to those who were married. Widowed status showed a trend towards higher odds but was not statistically significant (p = 0.057). Housing type also mattered, with those in permanent/semi-permanent housing having 1.385 times the odds (95% CI: 1.217–1.576, p < 0.001) compared to those in non-permanent structures. Household Ownership indicated that those in non-household owned dwellings had marginally significantly higher odds (AOR = 1.184, 95% CI: 1.000–1.403, p = 0.050) compared to owners. Interestingly, households experiencing Food Insecurity had significantly lower odds (AOR = 0.878, 95% CI: 0.782–0.985, p = 0.027) of unimproved water sources compared to food-secure households. Several other factors, including the age of the household head (across all categories), sex of the household head, school attendance, lighting source, IDP status, internet use, electricity access, and labour status, did not show statistically significant associations with the outcome unimproved drinking water sources in this adjusted model, as their p-values were greater than 0.05 and their 95% CIs for the AOR all included.

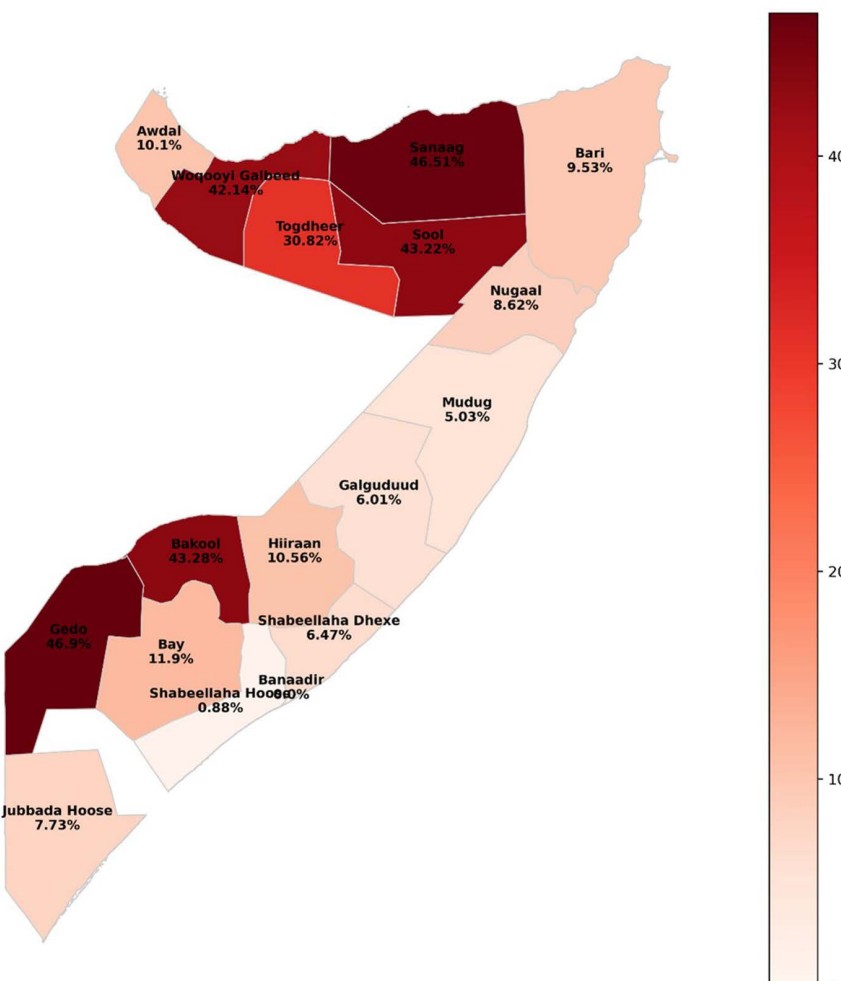

Distribution of Unimproved Water (%) in Somalia by Region

**Fig 1. Spatial Distribution of Poverty Magnitude among Somali Households (SIHBS 2022).**

## 4. Discussion

The findings highlight that reliance on unimproved water sources is a complex issue, significantly influenced by a confluence of community and individual/household factors, aligning with the multifaceted nature described in existing literature [7,8].

Community-factors, notably region and residence type, were strong predictors, consistent with studies in SSA and East Africa [9,10]. Significant regional disparities highlight localized challenges, likely reflecting varied infrastructure, hydrogeology, governance, and conflict [7,15]. Counterintuitively, urban households had higher odds of using unimproved water (AOR = 2.067) than rural ones, contrasting with general literature [5,7] and our bivariate results. This may stem from SIHBS definitions of "unimproved" (e.g., vendor-supplied water), the reference category, or unmeasured urban confounders. Nomadic populations showed no significant difference from rural ones post-adjustment.

At the household factors, marital status was significant, with households headed by divorced/never-married individuals more reliant on unimproved sources, possibly due to socioeconomic vulnerability [8,16]. Unexpectedly, permanent/semi-permanent housing was associated with higher odds of unimproved water use (AOR = 1.385), perhaps due to water

**Table 3. Multivariable Logistic Regression analysis of Factors Associated with Drinking Water Source.**

| Variable | Categories | AOR | Coefficient (S.E.) | 95% CI (AOR) | p-value |
|---|---|---|---|---|---|
| Age household head | <20 years | Ref | | | |
| | 20-29 years | 0.896 | −0.110 (0.0779) | 0.756 - 1.063 | 0.207 |
| | 30-39 years | 1.033 | 0.032 (0.1025) | 0.850 - 1.254 | 0.745 |
| | 40-49 years | 0.964 | −0.037 (0.1137) | 0.765 - 1.215 | 0.757 |
| | 50-59 years | 1.268 | 0.237 (0.1688) | 0.977 - 1.646 | 0.074 |
| | 60 + years | 1.109 | 0.103 (0.1325) | 0.877 - 1.402 | 0.387 |
| Sex of household head | Male | Ref | | | |
| | Female | 0.991 | −0.009 (0.0554) | 0.888 - 1.106 | 0.874 |
| School Attendance | Yes | Ref | | | |
| | No | 0.939 | −0.063 (0.0599) | 0.829 - 1.064 | 0.327 |
| Marital Status | Married | Ref | | | |
| | Divorced | 1.598 | 0.469 (0.2113) | 1.234 - 2.071 | 0.000 |
| | Never married | 1.262 | 0.233 (0.1079) | 1.067 - 1.492 | 0.006 |
| | Widowed | 1.247 | 0.221 (0.1450) | 0.993 - 1.567 | 0.057 |
| Housing type | Non-Permanent | Ref | | | |
| | Permanent/Semi-Permanent | 1.385 | 0.326 (0.0912) | 1.217 - 1.576 | 0.000 |
| Lighting source | Basic/None | Ref | | | |
| | Modern/Improved | 1.103 | 0.098 (0.1081) | 0.910 - 1.337 | 0.316 |
| Household Ownership | Yes | Ref | | | |
| | No, non-household | 1.184 | 0.169 (0.1023) | 1.000 - 1.403 | 0.050 |
| IDP | Yes | Ref | | | |
| | No | 0.820 | −0.199 (0.1073) | 0.634 - 1.059 | 0.129 |
| Internet use | Yes | Ref | | | |
| | No | 1.031 | 0.030 (0.0688) | 0.905 - 1.175 | 0.648 |
| Electricity Access | Yes | Ref | | | |
| | No | 1.144 | 0.134 (0.1177) | 0.935 - 1.399 | 0.192 |
| Residence | Rural | Ref | | | |
| | Urban | 2.067 | 0.726 (0.1415) | 1.807 - 2.364 | 0.000 |
| | Nomadic | 1.064 | 0.062 (0.1006) | 0.884 - 1.280 | 0.515 |
| Region | Banadir | Ref | | | |
| | Bakool | 0.165 | −1.800 (0.0259) | 0.122 - 0.225 | 0.000 |
| | Bari | 0.922 | −0.081 (0.1714) | 0.640 - 1.327 | 0.661 |
| | Bay | 0.994 | −0.006 (0.1826) | 0.694 - 1.425 | 0.975 |
| | Galgaduud | 1.780 | 0.576 (0.3581) | 1.200 - 2.640 | 0.004 |
| | Gedo | 0.118 | −2.136 (0.0185) | 0.087 - 0.161 | 0.000 |
| | Hiraan | 1.073 | 0.071 (0.1978) | 0.748 - 1.540 | 0.702 |
| | Lower Juba | 1.393 | 0.331 (0.3246) | 0.882 - 2.199 | 0.156 |
| | Lower Shabelle | 11.722 | 2.461 (5.4924) | 4.679 - 29.365 | 0.000 |
| | Waqooyi Galbeed | 0.139 | −1.973 (0.0199) | 0.105 - 0.184 | 0.000 |
| | Middle Shabelle | 1.652 | 0.502 (0.3391) | 1.105 - 2.470 | 0.014 |
| | Mudug | 2.081 | 0.733 (0.5060) | 1.292 - 3.352 | 0.003 |
| | Nugaal | 1.057 | 0.055 (0.1996) | 0.730 - 1.531 | 0.769 |
| | Sanaag | 0.109 | −2.219 (0.0170) | 0.080 - 0.148 | 0.000 |
| | Sool | 0.129 | −2.045 (0.0197) | 0.096 - 0.174 | 0.000 |
| | Togdheer | 0.247 | −1.397 (0.0383) | 0.183 - 0.335 | 0.000 |

*(Continued)*

**Table 3.** (Continued)

| Variable | Categories | AOR | Coefficient (S.E.) | 95% CI (AOR) | p-value |
|---|---|---|---|---|---|
| Labour status | Yes | Ref | | | |
| | No | 1.034 | 0.033 (0.0845) | 0.881 - 1.214 | 0.683 |
| Food Insecurity | Food Secure | Ref | | | |
| | Food Insecure | 0.878 | −0.130 (0.0516) | 0.782 - 0.985 | 0.027 |
| _cons | | 4.779 | 1.564 (1.2425) | 2.871 - 7.955 | 0.000 |

source classifications or lack of infrastructure connection despite better housing [7] (Aragaw et al., 2023). Non-ownership of dwellings showed marginally higher odds. Intriguingly, food-insecure households had *lower* odds (AOR = 0.878) of using unimproved water, a counterintuitive finding possibly reflecting resource prioritization, coping mechanisms, aid program links, or unmeasured confounders.

Several significant factors in the bivariate analysis, such as school attendance, lighting source, internet use, and electricity access, became non-significant in the multivariable model. This highlights the importance of multivariable analysis in controlling for confounding effects; these factors are likely correlated with stronger predictors like region or residence type, and their independent effect on water source choice is less pronounced once these major determinants are accounted for. The non-significance of the age and sex of the household head, IDP status, and labour status in the adjusted model suggests that, within the Somali context and after controlling for other factors, these characteristics are not primary drivers of reliance on unimproved water sources. This contrasts with some studies where education or sex of the household head showed significance [8,9], indicating context-specific variations.

In conclusion, tackling unimproved water access in Somalia demands strategies addressing regional disparities, urban supply dynamics, and household vulnerabilities. Counterintuitive findings, especially concerning urban residence and food insecurity, necessitate further investigation.

## 5. Limitations and future research

This study has several limitations. First, as a secondary analysis of the 2022 Somalia Integrated Household Budget Survey (SIHBS), the analysis was limited to variables collected in the original survey, and some potentially relevant factors, such as detailed household water infrastructure, water quality, and intra-household water use were unavailable. Second, the cross-sectional design precludes causal inference, and associations reported should not be interpreted as causal relationships. While multilevel logistic regression accounted for clustering of households within communities, spatially explicit techniques (e.g., geographically weighted regression) were not applied, which could have provided additional insights into localized disparities and geographic heterogeneity. Unmeasured confounding, particularly in urban and conflict-affected areas, may also partly explain some counterintuitive findings.

Despite these limitations, this study provides important nationally representative evidence on household- and community-level factors associated with unimproved drinking water use in Somalia. Future research could incorporate geocoded data and spatial modeling to explore geographic patterns, longitudinal designs to investigate causal pathways, and primary data collection on water quality and household practices to complement socioeconomic indicators. Additionally, examining intersectional vulnerabilities such as the combined effects of urbanization, socioeconomic status, and displacement would further inform context-specific interventions, supporting Somalia's progress toward Sustainable Development Goal 6.1.

## 6. Conclusion

This study used multivariable logistic regression analysis of the 2022 Somalia Integrated Household Budget Survey (SIHBS) to identify individual- and community-level factors associated with household reliance on unimproved drinking

water sources. Consistent with prior evidence, community-level factors, particularly region and residence type, emerged as strong predictors, underscoring substantial geographical disparities. Counterintuitively, urban households exhibited higher odds of using unimproved sources compared with rural households in the adjusted model, a finding that may reflect definitional issues within the SIHBS classification of "unimproved" sources or unmeasured urban dynamics. At the household level, marital status of the household head (divorced or never married) and, unexpectedly, permanent/semi-permanent housing were associated with greater reliance on unimproved water, while food-insecure households demonstrated lower odds. Importantly, several factors significant in the bivariate analysis, including education proxies, electricity access, and internet use were no longer significant once major predictors such as region and residence type were accounted for. This underscores the rationale for employing multivariable analysis: to control for confounding and assess the independent contribution of each factor after adjusting for correlated variables. Variable selection was informed by both theoretical relevance, as established in prior literature, and empirical evidence from descriptive and bivariate analyses. While the cross-sectional nature of the data precludes causal inference, the associations identified here provide important insights into context-specific vulnerabilities and inequalities in water access. These findings support the design of geographically targeted and socially responsive interventions, while highlighting areas, such as unexpected urban and food insecurity patterns that warrant deeper investigation through longitudinal or mixed-methods research.

## Supporting information

**S1. Custom code used for the analysis of the Spatial Distribution of Poverty Magnitude among Somali Households (SIHBS 2022).**
(PDF)

## Author contributions

**Conceptualization:** Omar Muhumed Maidhane.

**Data curation:** Omar Muhumed Maidhane.

**Formal analysis:** Omar Muhumed Maidhane.

**Methodology:** Omar Muhumed Maidhane.

**Resources:** Omar Muhumed Maidhane.

**Software:** Omar Muhumed Maidhane.

**Supervision:** Omran Salih.

**Validation:** Omar Muhumed Maidhane.

**Visualization:** Omar Muhumed Maidhane.

**Writing – review & editing:** Omran Salih.

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
