## [Decision Letter · Decision Letter 0]

4 Aug 2025

Dear Dr. Salih,

We look forward to receiving your revised manuscript.

Kind regards,

Khin Thet Wai, MBBS, MPH, MA

Academic Editor

PLOS ONE

Journal Requirements: 

Additional Editor Comments:

Extensive major revisions are required for methods and results particularly data analysis, interpretation of multivariate analysis and appropriate discussion and conclusions.

Reviewers' comments:

Reviewer's Responses to Questions

**Comments to the Author**

1. Is the manuscript technically sound, and do the data support the conclusions?

Reviewer #1: Yes

Reviewer #2: No

2. Has the statistical analysis been performed appropriately and rigorously?

Reviewer #1: Yes

Reviewer #2: No

3. Have the authors made all data underlying the findings in their manuscript fully available?

Reviewer #1: Yes

Reviewer #2: Yes

4. Is the manuscript presented in an intelligible fashion and written in standard English?

Reviewer #1: No

Reviewer #2: Yes

Reviewer #1: 1.Rewrite the abstract in a proper structure (e.g., background, methods, results, conclusion).

2.Provide a more precise rationale for using a multivariable analysis.

3.The study uses cross-sectional data but occasionally provides causal conclusions. Causal conclusions cannot be validly drawn from such dataAuthors need to provide a clear description about reasons for selecting variables in the multivariable model.

4.Some AORs (e.g., for Lower Shabelle) have extremely wide CIs, suggesting sparse data or overfitting. This should be admitted.

5.The variable “food insecurity” shows a negative association with unimproved water use. It contradicts our expectation. Similar is the case with urban households. Authors can provide plausible reason behind it.

6.Authors can provide figures/maps to show regional variations.

7.Acknowledge the limitations of secondary data.

8.The use of terms like "community-factors factors" should be fixed throughout.

9.Several grammatical issues should be corrected.

Reviewer #2: Study presented in the manuscript is interesting, which provides a better understanding of drinking water sources and associated factors in Somalia. However, this study lacks novelty. Using of logistic regression for examining the association between various variables is very common and been used for similar type of studies before.

This study aims at investigate the prevalence of unimproved drinking water source use and identify its associated individual and community factors among households in Somalia using household survey data. As this data was collected at household level, identifying geographically targeted interventions that address regional inequalities is important. Though the authors mentioned about geographical disparity in the abstract, it is not reflected in the study.

Using of any spatial data analysis technique modeling spatial relationships, such as geographically weighted regression could provide a better understanding of the drinking water availability scenario in Somalia.

Moreover, results provided in tables 1 and 2 are a bit monotonous. These tables could be divided into smaller tables depending on variable type.

Maps generated through spatial analysis techniques could represent the household survey data in a much better way.

**Do you want your identity to be public for this peer review?** For information about this choice, including consent withdrawal, please see our Privacy Policy

Reviewer #1: No

Reviewer #2: **Yes: ** Chandan Roy

---

## [Author Response · Author response to Decision Letter 1]

18 Sep 2025

Manuscript ID: PONE-D-25-33507

Manuscript Title: Unimproved Source of Drinking Water and Its Associated Factors: Multivariable Analysis of Somalia Integrated Household Budget Survey (SIHBS 2022)

To: PLOS ONE

Re: Response to reviewers

Dear Respected Editor,

Thank you for allowing a resubmission of our manuscript, with an opportunity to address the reviewers’ comments.

We are uploading (a) our point-by-point response to the comments (below) (response to reviewers), (b) an updated manuscript with yellow highlighting indicating changes, and (c) a clean updated manuscript without highlights.

Finally, we would like to thank the reviewers for reviewing the paper and providing suggestions to improve the paper’s quality. The authors have addressed the comments and suggestions, as described in the response below.

We look forward to your decision.

Best regards,

The paper’s authors

Reviewer #1

Reviewer #1, Concern #1: Rewrite the abstract in a proper structure

Authors response, Concern #1: We appreciate your time and effort in reviewing our work. We have carefully considered your comments and have made the suggested revisions in the updated version. (See Page 1, Abstract).

Reviewer #1, Concern #2: Provide a more precise rationale for using a multivariable analysis.

Authors response, Concern #2: We have revised the manuscript based on the reviewer's concerns to precise rationale using multivariable analysis (See Page 6, Section 2.6).

Reviewer #1, Concern #3: The study uses cross-sectional data but occasionally provides causal conclusions. Causal conclusions cannot be validly drawn from such data Authors need to provide a clear description about reasons for selecting variables in the multivariable model.

Authors response, Concern #3: We have revised the Conclusion section to ensure that our interpretation avoids causal language and is framed strictly in terms of associations. We now explicitly acknowledge the cross-sectional nature of the SIHBS 2022 data and the consequent limitations in drawing causal inferences (See Page 18, Section 6).

Reviewer #1, Concern #5: Some AORs (e.g., for Lower Shabelle) have extremely wide CIs, suggesting sparse data or overfitting. This should be admitted.

Authors response, Concern #5: Thank you for your insightful comments, when compared to a near-zero reference, can lead to extremely high AORs with wide CIs. The reference group (Banadir) has zero unimproved water users in the sample, which makes calculating an odds ratio against it problematic.

Reviewer #1, Concern #6: The variable “food insecurity” shows a negative association with unimproved water use. It contradicts our expectation. Similar is the case with urban households. Authors can provide plausible reason behind it.

Authors response, Concern #6: Thank you for your insightful comments and for highlighting these two crucial points regarding the unexpected associations between urban residence and food insecurity with the use of unimproved drinking water. We agree that these findings initially appear counterintuitive and represent key contributions of our multivariable analysis. We appreciate the opportunity to elaborate further on the plausible reasons behind these observations, as discussed in our manuscript's Discussion section (Page 18, Section 4).

Reviewer #1, Concern #7: Authors can provide figures/maps to show regional variations.

Authors response, Concern #7: we have provided a figure/map to show regional disparities (See page 12, Figure 1).

Reviewer #1, Concern #8: Acknowledge the limitations of secondary data.

Authors response, Concern #8: Thank you for your insightful comments, we considered the limitations of secondary data (see page 4, section 2.3)

Reviewer #1, Concern #9: The use of terms like "community-factors factors" should be fixed throughout.

Authors response, Concern #9: We have considered the reviewer's concern in the revised manuscript.

Reviewer #1, Concern #10: Several grammatical issues should be corrected.

Authors response, Concern #10: We have considered the reviewer's concern in the revised manuscript.

Reviewer #2:

Reviewer #2, Concern #1: Study presented in the manuscript is interesting, which provides a better understanding of drinking water sources and associated factors in Somalia. However, this study lacks novelty. Using logistic regression for examining the association between various variables is very common and been used for similar type of studies before.

Authors response #2, Concern #1. We sincerely thank the reviewer for this observation. We agree that logistic regression is a widely used method in public health research. However, the novelty of our study lies not solely in the choice of statistical technique, but in the context, dataset, and findings. Specifically:

1. First use of SIHBS 2022: This is the first study to conduct a comprehensive multivariable analysis of the most recent and nationally representative Somalia Integrated Household Budget Survey (SIHBS 2022). Given the limited availability of reliable household-level data in fragile and conflict-affected contexts, this represents a unique opportunity to generate robust, up-to-date evidence on water access in Somalia.

2. Context-specific insights: Our study provides the first nationally representative evidence on drinking water access in Somalia in the context of ongoing conflict, displacement, and climate change factors that distinguish Somalia from many settings previously studied.

3. Identification of counterintuitive associations: Beyond confirming known patterns, we identify surprising and context-specific findings (e.g., urban households exhibiting higher odds of using unimproved water sources and food-insecure households showing lower odds), which challenge assumptions established in broader Sub-Saharan African studies and highlight complexities unique to Somalia.

4. Policy relevance: By highlighting regional disparities and the interaction of household vulnerabilities with water access, our findings provide actionable, context-specific evidence to inform Somali authorities and international partners working towards Sustainable Development Goal 6.1.

Therefore, while logistic regression itself is not novel, the application of this method to the SIHBS 2022 dataset in Somalia, the unique context of fragility and climate vulnerability, and the counterintuitive findings generated collectively establish the originality and contribution of our study. We have also added a dedicated Novelty of the Study (Section 2.7) to clearly articulate these contributions in the revised manuscript.

Reviewer #2, Concern #2: This study aims at investigate the prevalence of unimproved drinking water source use and identify its associated individual and community factors among households in Somalia using household survey data. As this data was collected at household level, identifying geographically targeted interventions that address regional inequalities is important. Though the authors mentioned about geographical disparity in the abstract, it is not reflected in the study.

Authors response #2, Concern #2 Thank you for your insightful comments. We considered identifying geographically targeted interventions that address regional inequalities and a figure/map is provided to show regional disparities (See page 12, Figure 1).

Reviewer #2, Concern #3: Using of any spatial data analysis technique modeling spatial relationships, such as geographically weighted regression could provide a better understanding of the drinking water availability scenario in Somalia.

Authors response, Concern #3. We greatly appreciate this insightful suggestion. We fully agree that spatial analysis techniques, such as geographically weighted regression (GWR), can provide valuable insights into geographic heterogeneity and spatial dependence in water access. However, the primary objective of the present study was to conduct a nationally representative multivariable analysis of household- and community-level factors associated with unimproved drinking water use, using the SIHBS 2022 survey design. Given the structure of the dataset and the focus on nested household–community relationships, we employed multilevel logistic regression, which allowed us to appropriately account for clustering, estimate intraclass correlation, and disentangle individual- and community-level effects.

Nonetheless, we acknowledge that spatial techniques like GWR could enrich the analysis by identifying localized variations in determinants of water access that may not be fully captured in a multilevel framework. We have added a note in the Limitations and Future Research section to highlight that future studies should explore spatially explicit models, where geocoded data are available, to complement our findings and provide a deeper understanding of spatial disparities in Somalia.

Reviewer #2, Concern #4: Moreover, results provided in table 1 are a bit monotonous. The table could be divided into smaller tables depending on variable type.

Authors response, Concern #4. We considered restructuring these tables by separating them according to variable type (See pages 11-12, Tables 1 and 2).

Reviewer #2, Concern #5: Maps generated through spatial analysis techniques could represent the household survey data in a much better way

Authors response, Concern #5. Thank you for your insightful comments, we have provided a figure/map to show regional disparities (See page 12, Figure 1).

We sincerely thank the reviewers for their constructive comments and suggestions, which have helped us improve the clarity, rigor, and contribution of our manuscript.

---

## [Decision Letter · Decision Letter 1]

30 Sep 2025

Unimproved Source of Drinking Water and Its Associated Factors: Multivariable Analysis of Somalia Integrated Household Budget Survey (SIHBS 2022)

PONE-D-25-33507R1

Dear Dr. Salih,

We’re pleased to inform you that your manuscript has been judged scientifically suitable for publication and will be formally accepted for publication once it meets all outstanding technical requirements.

Kind regards,

Khin Thet Wai, MBBS, MPH, MA

Academic Editor

PLOS ONE

Additional Editor Comments (optional):

All comments are adequately addressed.

Reviewers' comments:

Reviewer's Responses to Questions

**Comments to the Author**

Reviewer #1: All comments have been addressed

Reviewer #2: All comments have been addressed

2. Is the manuscript technically sound, and do the data support the conclusions?

Reviewer #1: Yes

Reviewer #2: Yes

3. Has the statistical analysis been performed appropriately and rigorously?

Reviewer #1: Yes

Reviewer #2: Yes

4. Have the authors made all data underlying the findings in their manuscript fully available?

Reviewer #1: Yes

Reviewer #2: Yes

5. Is the manuscript presented in an intelligible fashion and written in standard English?

Reviewer #1: Yes

Reviewer #2: Yes

Reviewer #1: The authors have addressed my concerns. However, minor errors such as spelling mistakes, typos, and grammatical issues should be carefully checked before publication.

Reviewer #2: (No Response)

**Do you want your identity to be public for this peer review?** For information about this choice, including consent withdrawal, please see our Privacy Policy

Reviewer #1: No

Reviewer #2: **Yes: ** Dr. Chandan Roy, Professor, Department of Geography and Environmental Studies, University of Rajshahi, Rajshahi-6205, Bangladesh.

---

## [Editor Report · Acceptance letter]

PONE-D-25-33507R1

PLOS ONE

Dear Dr. Salih,

I'm pleased to inform you that your manuscript has been deemed suitable for publication in PLOS ONE. Congratulations! Your manuscript is now being handed over to our production team.

Kind regards,

on behalf of

Dr. Khin Thet Wai

Academic Editor

PLOS ONE